# Mixed Reality for Safe and Reliable Human-Robot Collaboration in Timber Frame Construction

**Luis Felipe González-Böhme and Eduardo Valenzuela-Astudillo \***

Department of Architecture, Universidad Técnica Federico Santa María, Valparaíso 2390123, Chile; luisfelipe.gonzalez@usm.cl
* Correspondence: eduardo.valenzuelaa@usm.cl

**Abstract:** In the field of construction, human-robot collaboration and mixed reality (MR) open new possibilities. However, safety and reliability issues persist. The lack of flexibility and adaptability in current preprogrammed systems hampers real-time human-robot collaboration. A key gap in this area lies in the ability of the robot to interpret and accurately execute operations based on the real-time visual instructions and restrictions provided by the human collaborator and the working environment. This paper focuses on an MR-based human-robot collaboration method through visual feedback from a vision-based collaborative industrial robot system for use in wood stereotomy which we are developing. This method is applied to an alternating workflow in which a skilled carpenter lays out the joinery on the workpiece, and the robot cuts it. Cutting operations are instructed to the robot only through lines and conventional "carpenter's marks", which are drawn on the timbers by the carpenter. The robot system's accuracy in locating and interpreting marks as cutting operations is evaluated by automatically constructing a 3D model of the cut shape from the vision system data. A digital twin of the robot allows the carpenter to previsualize all motions that are required by the robot for task validation and to know when to enter the collaborative workspace. Our experimental results offer some insights into human-robot communication requirements for collaborative robot system applications in timber frame construction.

**Keywords:** mixed reality; human-robot collaboration; digital twin; robotic manufacturing; digital wood stereotomy; timber frame construction

## 1. Introduction

According to ISO/TS 15066:2016 [1], collaborative operation is the state in which a purposely designed industrial robot system and a human operator work within a collaborative workspace. More explicitly, in collaborative robot operations, operators can work in close proximity to the robot system while the power to the robot's actuators is available, and physical contact between an operator and the robot system can occur. A collaborative workspace is a space within the operating space where the robot system—including the workpiece—and a human can perform tasks concurrently during production operations [1]. Despite the numerous implications and complexities that are inherent in close-proximity human-robot collaboration, this standard notably lacks provisions concerning communication or feedback between the robotic system and the operator.

When it comes to industrial robots, their reliability depends on several key factors, such as monitoring and collecting data about robot behavior, failures, calibration, end-effectors, and following the manufacturer's maintenance recommendations [2]. Moreover, when considering the field of human-robot collaboration, establishing reliable communication channels between the operator and robot is critical. These channels not only facilitate the accurate communication of work instructions but also play an important role in ensuring safety [3,4]. In the dynamic environment of a collaborative workspace, robust communication systems should provide real-time feedback, allowing operators to react

promptly to any unexpected robot behavior or potential safety hazards, thereby minimizing risk and maximizing productivity [5,6].

The layout is the highly skilled process of locating and marking stereotomic shapes to connect the timbers in a timber frame. Cutting is the execution of the layout, which mainly involves sawing, drilling, chiseling, and planning. Due to the level of skill involved, more senior carpenters tend to perform most of the layout, but when it comes to cutting the joints, everyone pitches in [7]. This process establishes an efficient communication method between carpenters. Graphic language consisting of lines and symbols to indicate the location and orientation of timbers in the frame to the location and shape of the joinery allows carpenters from the same team—or the same geographic region—to exchange layout and cutting tasks.

Carpenters' methods adapt to the variations in shape and the dimension of each workpiece. Green, rough-sawn, or hand-hewn timbers are no problem for a skilled carpenter. Neither are logs or twisted tree shapes. Carpenters adapt their workspace to specific job site conditions and choose the right tool for the job when faced with the task. In contrast, CAD/CAM/CNC methods require calibrated, kiln-dried lumber, and a fixed workspace to achieve the best quality results. In addition, the operator must know the proprietary programming language. Human adaptability is hard to automate and is valuable in many domains, most notably in real-world problem-solving and creative task performance [8].

We believe that the potential threat of automation to traditional skilled trades such as carpentry can be overcome by adapting robots to humans. The methodological differences between traditional carpenters and robotic manufacturing have been considered, raising the challenge of designing an appropriate communication system for human-robot collaboration in timber frame construction. Given the user's limited specialized knowledge in advanced manufacturing with industrial or collaborative robots, it becomes critical to develop a communication strategy that demystifies and elucidates the essential principles of robotic operations in the context of traditional woodworking. To this end, we propose maintaining the existing layout techniques with minimal modifications, thus requiring the robotic system to adapt accordingly.

Considering the safety-rated monitored stop method (ISO/TS 15066:2016 [1]) of collaborative operation, we developed a system that is capable of displaying robot movement given a specific task and the manufacturing result, responding to various workpiece configurations for timber manufacturing, and expressing the collaborative workspace dimensions. Within this, mixed reality visual feedback enables the operator to directly observe in advance robotic motion and any hazards that might arise from this motion, including either possible collisions with the operator or elements of the working environment.

The results of this investigation show that this interfacing-translation method can deliver an intuitive and reliable way to display the needed information and interactions for human-robot collaboration in timber construction, giving enough precision for the positioning of digital elements in the real environment, the effective translation of the cutting resulting from symbols in the layout of the workpiece, expressing robot behavior, and thus developing reliability and safety awareness.

These findings establish important knowledge about how the use of mixed–reality interfaces can contribute to explaining the effect of advanced manufacturing processes in the human-robot shared space, both as a formative and fast integration instance, as well as to speed up decision-making in human-robot collaborative environments; these are, along with flexibility and dexterity, some of its key contributions compared to fully automated instances. Through ongoing research, it is expected that the implications of this method will be further explored with the testing of a wider variety of timber construction tasks.

## 2. Literature Review

### 2.1. Human-Robot Collaboration in Wood Stereotomy

Small and medium-sized manufacturers are increasingly interested in the potential productivity benefits of combining the creativity and ability of humans to solve ill-structured

problems with the strength and repeatability of the industrial robot but there are a variety of economic and technical factors that limit their adoption [9,10]. Human-robotic collaboration frameworks are mentioned as an alternative and an opportunity, that offers a significant contribution to manufacturing flexibility versus rigidity but with the greater efficiency of fully automated systems [11–13]. It is argued that to succeed in the Industry 4.0 era, workers need to acquire a wide range of specific skills, facing the need to combine conventional knowledge with computer skills [12], while research is developing new ways of communicating with cyber-physical systems that allow efficient integration, considering factors such as reliability, user-friendliness, security, and productivity [14].

Although collaborative robotics is implemented in several industries, in wood prefabrication tasks, the solutions are still scarce. The work conducted by the CREATE research group at the University of Southern Denmark—in which SDU stands out through developing solutions for wood structure assembly with a focus on end-effector control and guidance [15–17]—designing structured workspaces has demonstrated successful collaborative working executions implementing manual guidance; however, this still depends on exhaustive programming procedures specific to the results preconfigured and without further elaboration on safety issues, beyond the inspection of previous simulations that allow a safety distance to be approximated and verifying that the instructions have been entered correctly.

References [18–20] have addressed HRC in manufacturing processes using hand drawings on the workpiece to communicate the toolpath to an industrial robot. Pedersen et al.'s research [20] bears the closest resemblance to our work so far. They put forth a method for robotic fabrication that used parametric visual feedback to identify hand-drawn markings on a given object. Employing a camera, their system was designed to spot either open/closed curves or lines indicative of intended robotic cutting paths. While innovative in its design, its methodology presents a limitation in only accommodating two kinds of operations, thus not providing a sufficiently diverse language to enable a variety of operations on the workpiece.

Our proposed system does not directly communicate the toolpath in the drawings on the workpiece. The symbol language provides instructions regarding subtractive manufacturing procedures to be interpreted for the automated generation of tool paths, reaching the designed geometry. The symbol's language is expressed on different faces of the workpiece; these symbols must be linked for the three-dimensional interpretation of a target geometry, e.g., indicating the dimensions of a mortise milling with a rectangular profile on one face and the depth and angle of cut on another.

### 2.2. Safety-Aware Human-Robot Collaboration: Visual Solutions

One of the main concerns in the close collaboration between humans and robots has been worker safety [3,21]. Safety in HRC raises challenging requirements as humans work near robots without fences or guard cells. Various safety strategies are still being developed to ensure collaboration and productivity [14,22–24], usually by assessing and restricting the speed and distance between humans and robots and taking into consideration that each manufacturing plant may have different configurations for collaborative workspaces (Figure 1).

Researchers have taken different approaches to developing collision-free human-robot collaboration (HRC) systems. One approach is based on context awareness, where the system can plan robotic paths that avoid colliding with human operators while still reaching target positions in time [13], providing both human safety and assembly efficiency. Another approach is to use sensor-based safety systems, where the distance between human operators and robots can be actively monitored, and robots can be controlled to stop or move away if the distance between them is too short, which can significantly increase the time of the collaborative manufacturing process.

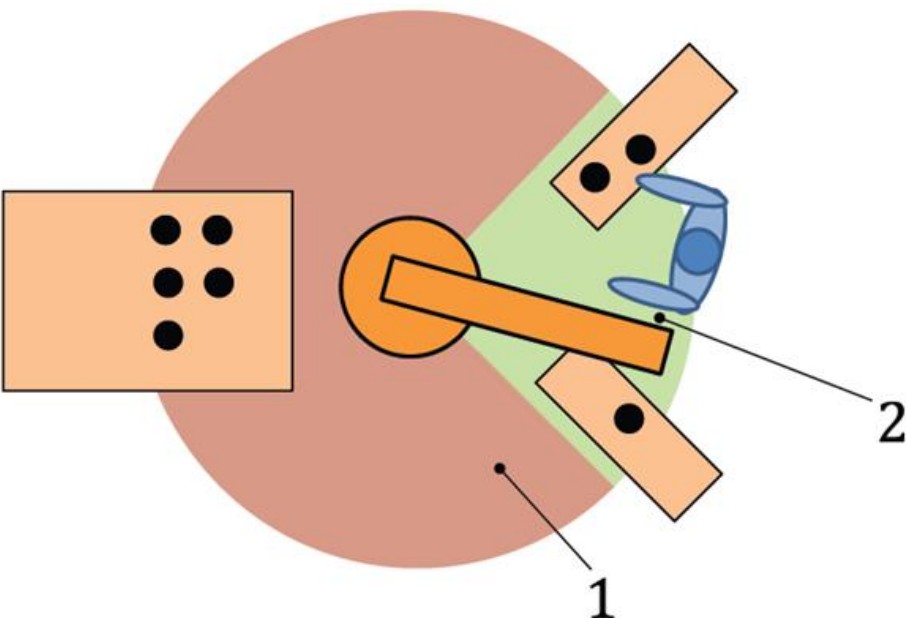

**Figure 1.** Example of a collaborative workspace: (1) Operation space; (2) Collaborative workspace, according to ISO/TS 15066:2016 [1].

Additionally, deep learning models using two-dimensional images have become viable solutions for human motion detection along with artificial intelligence techniques. Sajedi et al. [14] have suggested a probabilistic interpretation method and a framework for utilizing a deep model's uncertainty to increase the accuracy of HRC's human motion detection.

Visual solutions, which guide the HRC operator, can establish a significant contribution to safety awareness and injury avoidance. Graphically displaying the movement of the robot in the workspace prior to execution as safety information can help avoid collisions between the robot, the operator, and the working environment (e.g., worktable, workpieces, and other tools). A combination of real-time monitoring, digital twins [25], and head-mounted displays have proven promising in discussions and results.

Currently developing solutions, such as the work presented by Choi [24], propose an integrated system with sensors capable of perceiving the user's position and pose in real time, allowing for a more accurate assessment of safety issues in relation to a manufacturing process, this is applicable in the context of collaborative robotics for wood prefabrication work, where it becomes necessary to evaluate spatial constraints of greater complexity and the development of solutions adaptable to environments without previous structuring, with cumulative procedures that can be reported with a greater variety of data compared to what is presented in this reference.

Tehrani et al. [26] proposed a framework for mapping work zones of human-robot collaboration in a fenceless mixed reality environment, using machine learning prediction models and virtual reality.

Our ongoing research focuses on human-robot communication for wood stereotomy tasks in unstructured environments. Since a hand-drawn instruction system is implemented for the robot, it is essential that the system allows an agile and localized inspection of the work environment, which allows an evaluation if the instruction entered is appropriate and does not present risks for the operator, evaluating its position in the working environment.

### 2.3. Mixed Reality Assisted Manufacturing in HRC

It has been demonstrated that mixed reality with a head-mounted display or smart glass can make a significant contribution to performance in manufacturing tasks [27], positioning working instructions by means of holograms in the workspace. Nevertheless,

there are few studies on its use in carpentry tasks, especially in unstructured environments for timber joinery.

The need for connectivity and interactivity with digital information that has direct implications in the real environment is stronger than ever in the face of the increased use of cyber-physical systems, the use of digital twins, and artificial intelligence, among others [28]. A valuable particularity of MR for the purpose of the present project lies in its ability to deliver aggregated information in the workspace that is immediately linked to the elements on site, which allows help for unskilled workers to accelerate and strengthen their training [29]. It is possible to bet on a manufacturing model with digital twins that link and communicate in real time the implications of the process, in advance, for decision making and safety. In terms of integration, as proposed, the possibility of streamlining the representation of information needed to perform carpentry tasks holds a relevant and still little-explored contribution in this field.

Facing the problem of verifying whether the robot correctly understands the instructions given by the human with alternative methods to offline programming, mixed reality techniques make a significant contribution by allowing a preview of the robot's movement directly in the workspace and eventual transformations in the workpiece or in a composition. This has been demonstrated by some works [30,31], which involve the presence of digital twins expressed as holograms through a mixed-reality viewer positioned co-incidentally with the real robot. Usually, these interfaces allow the preview of the robot displacement and execution control but do not display relevant information regarding the operation space and collaborative workspace defined in applicable standards (ISO/TS 15066:2016 [1]); it is still necessary to express data (visual guides) that allow an evaluation of the in situ possibilities and restrictions of collaborative work in various spatial configurations, both predictively and with sufficient fidelity, as an input for the consolidation of a safety strategy.

Specifically in the field of human-robot collaboration and timber construction, examples are still scarce. Of note is the work conducted by Kyjanek [30], who managed to prototype a system that allowed controlling operations of a robotic manipulator in a preconfigured environment for assembly tasks, previewing trajectories.

Since we have considered highly skilled carpenters as users, problems such as the deficit in spatial thinking, an issue commonly studied in mixed reality developments (e.g., [32]), is not addressed in the research, although contributions in this field are recognized by allowing an explicit representation of the geometric result of the layout interpretation performed by the system. It is considered that the operator has the necessary competencies and skills to understand the woodworking layout, fully interpreting the geometric implications of the drawings on the workpiece, but not the specific result of the procedure developed by an industrial robot in collaboration and the adjustments that may involve the use of some tools such as spindles in the replacement of traditional tools, such as chisels and saws. Eventually, however, it is possible that these could be incorporated as end effectors of a robot at some point in time. The focus of attention lies in the feedback that a mixed reality system can provide regarding the displacement and space required by an industrial robot in collaborative work.

## 3. Materials and Methods

This ongoing research proposes an integrated MR system for safety-aware HRC in wood stereotomy using a vision-based collaborative industrial robot system with visual feedback and digital twin and sensor-based user recognition for a real-time safety collision evaluation between the robot and human operator.

As stated in previous sections, the purpose of this work involves adapting robotic manufacturing methods to the traditional way that carpenters work. For this reason, the workflow in the proposed method starts with the layout process in the workpiece, which acts as an instructional visual language for the robot; then, the mixed reality assistance contributes to human-robot communication, expressing visual feedback on the cutting

process in advance for safety awareness and decision making on the manufacturing process (Figure 2).

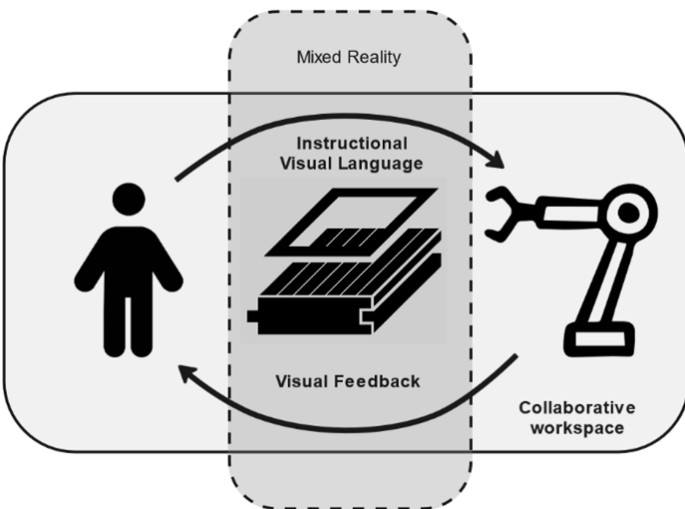

**Figure 2.** Mixed reality communication system for timber composite construction.

Our ongoing research has established a hand-drawn language close to carpenters' symbology, potentially easing the use of collaborating with robots on manufacturing and reducing the chances of frustration and lower adoption [33]. A 3D model of the resulting cut shape is automatically constructed from the data read by a vision system, and tool paths are calculated for the robot, based on preconfigured methods for wood stereotomy, such as mortise and tenon or dovetail joints, among others. The same information allows us to set the behavior of a digital twin for the robot, delivering manufacturing and safety information in the form of visual guides and virtual 3D models projected within a mixed reality head-mounted display (MR-HMD).

Our solution is executed through a sequential four-step process which includes layout drawing on the workpiece, the positioning and scanning of the workpiece for automatic detection and comprehension, mixed-reality visual feedback, and finally, the execution of the given instruction (Figure 3).

### 3.1. Experimental Setup

An experimental setup was established to ease the development and testing of this method while also providing a feasible working environment for wood stereotomy. This system allowed us to preview and analyze the trajectory, movement, and orientation of the tool commanded from hand drawings on the workpiece.

The equipment used in the workbench for proof-of-concept of the proposed method consists of 4 primary devices: (1) a UR5 collaborative robot, (2) a Zivid Two stereo camera, (3) a steel scriber as an end-effector and TCP marker, (4) a Microsoft HoloLens 2 MR-HMD, and (5) a computer with a Windows 10 operating system (Figure 4).

### 3.2. Layout: An Instructional Visual Language for HRC

The timber joinery layout process consists of a series of symbols established in ongoing research developed by our team (Figure 5). This research emphasizes the problem of detecting the drawings with computer vision and its identification, satisfying self-imposed constraints regarding similarity in the drawn language of the carpenters and the easy recognition of the symbols using a mounted camera in a collaborative robot, with computer vision and machine learning techniques.

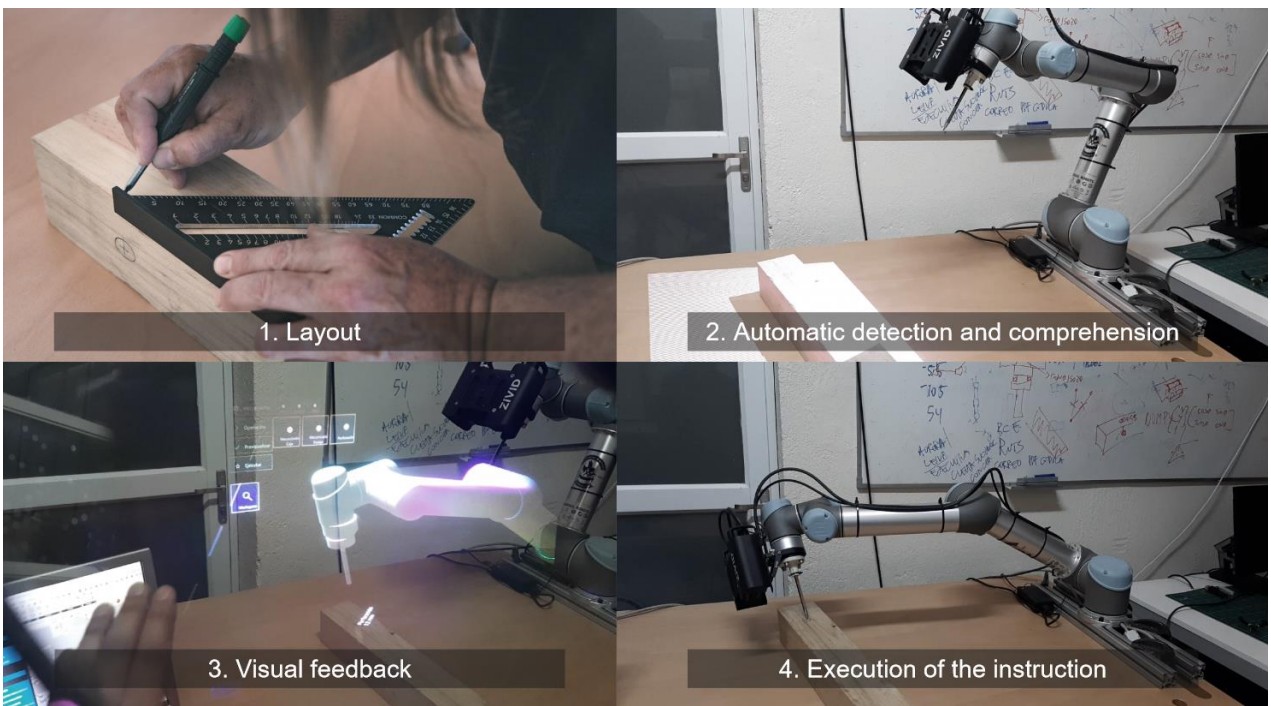

**Figure 3.** MR-based human-robot workflow for timber construction.

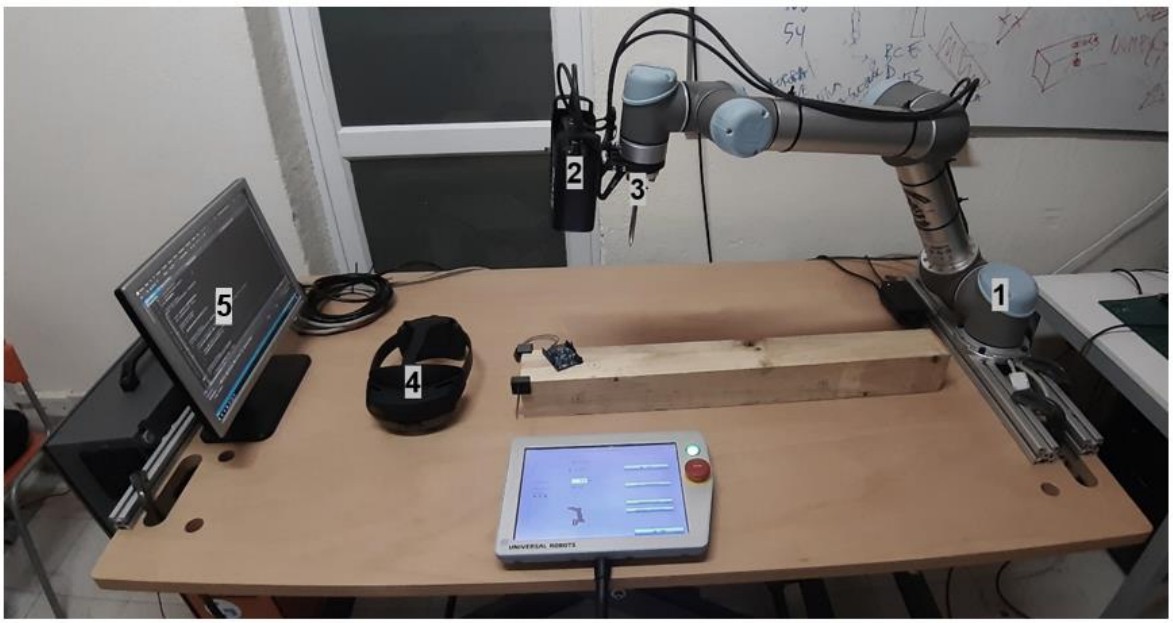

**Figure 4.** Proof-of-concept setup.

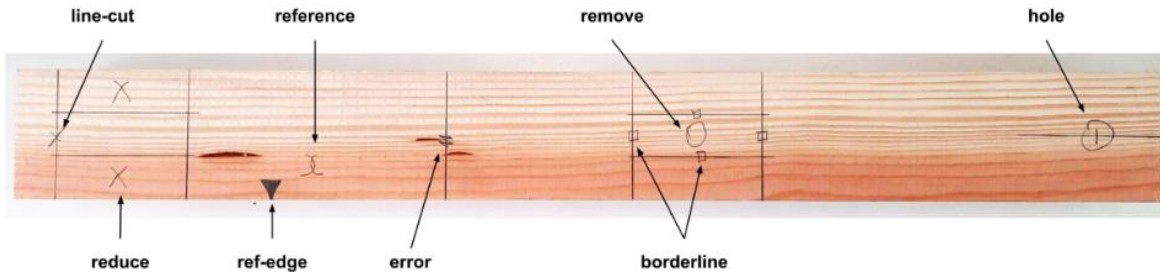

**Figure 5.** Example of timber joinery layout using our proposed hand-drawn language (ongoing research).

Using a ZIVID 2 depth camera (Oslo, Norway), the workpiece with the hand-drawn layout was scanned to obtain a mapped workpiece in 3D, giving positional data in relation to the robot's base coordinate system (Figure 6). After that, symbol detection started using a trained CNN-based object detector (YOLOv5). This information is then processed and transferred to a dedicated module programmed in Grasshopper, a visual programming language for parametric design within the Rhino 7 CAD environment, using primarily the Robots v.1.5.2 plugin [34] to create and simulate robotic programs. Thus, the position, orientation, and type of instruction could be recognized and used as input in a toolpath planning and safety feedback system, which later allowed information to be displayed on the workpiece within the MR-HMD.

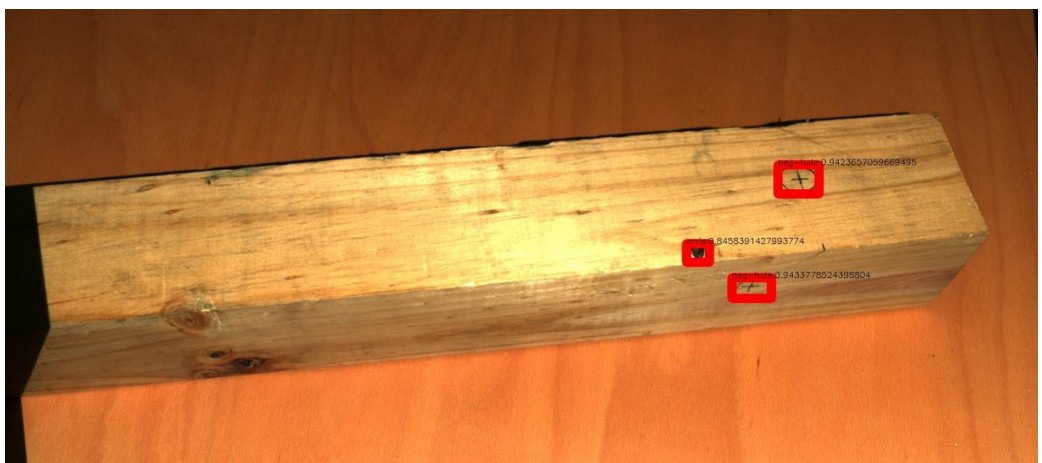

**Figure 6.** Drilling instruction reading prototype; camera perspective.

It was expected that the inspection of the workpiece with the layout drawn by the operator could provide information on points, curves, or surface regions since it is needed to set targets for the collaborative robot. The grasshopper module uses this information as input for a given operation that queries a preconfigured library of manufacturing processes created by us. This library includes algorithms for the automatic generation of tool paths like various Computer-Aided Manufacturing (CAM) programs have. In the context of the current experimental process, basic values such as position, orientation, width, or depth were required. It is worth noting that different operations may require other types of information.

In this research, we focused on visual feedback and safety awareness in human-robot collaboration (HRC). To assess the system's behavior, we used simulated data points (x, y, z) representing the layout symbols and workpiece scanning results (Figure 7). While these points were manually incorporated in the current phase, in future development, they could be directly and automatically derived from the computer vision system. This methodology enabled us to construct MR visual feedback and test toolpath curves for drilling, mortise, and tenon joinery and sanding sequences on a surface.

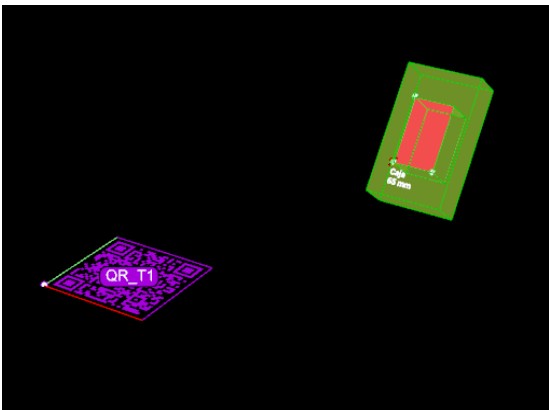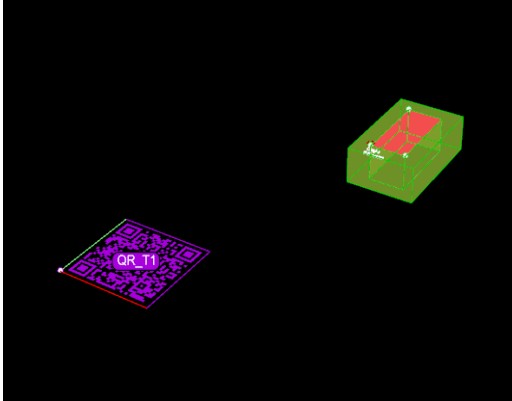

**Figure 7.** Preliminary visual feedback construction and positioning example with 3 input points from the workpiece and mortice depth parameters.

The virtual data allowed us to evaluate the performance and effectiveness of the proposed experimental cyber–physical system in various tasks and scenarios along the working space. Based on these data, all the geometry for 2D and 3D visual feedback (robot and workpiece modeling), toolpaths, and robot programming were developed in Grasshopper through the integration of a series of plugins and resources that informed and enhanced the system (Figure 8).

The toolpaths and the robot movement were then executed and evaluated using a steel scriber as the end effector for position and orientation verification. Subsequently, the same operations were emulated using virtual tools, including an electric spindle with a flat-end milling cutter and an orbital sander.

The system we are proposing relies on pre-existing 3D models of tools designed for a UR5 robot. By utilizing inverse kinematic calculations conducted in the robot programming software, the pose of the robot could be accurately determined. This approach bypasses the need for creating tool models through 3D scanning: a method commonly employed in similar research as per the literature review. This is because the position, orientation, shape, and size of the end effectors are known. This integration allows for a proactive assessment of how the robot approaches the workpiece, warning about robotic errors and preventing potential collisions of the robot with itself, the operator, or the working environment.

### 3.3. Mixed–Reality Interface

As a method to show the comprehension made by the system from the drawings that the carpenter makes on the workpiece (the layout), as well as the movements and workspace used by the robot, a Microsoft Hololens 2 (Redmond, WA, USA) mixed reality head-mounted display device (MR-HMD) was used, taking advantage of built-in computer vision algorithms, such as SLAM (simultaneous localization and mapping) to obtain the movement of the device, as well as the spatial mapping algorithms to obtain 3D meshes of the environment, archiving accurate positioning.

Real-time, three-dimensional data regarding the robot's movement made in Grasshopper were transmitted to a custom program developed within Unity 2020.3.30f with XR SDK and a Microsoft Mixed Reality Toolkit 2.7.1 (MRTK) using the Rhino. Inside system. This program employs a specialized algorithm that aligns the digital twin of the robot with its real-world counterpart, establishing anchors in the real space. In a similar vein, the geometric outcome of the intended operation is projected onto the workpiece. Data pertaining to dimensions and the specifics of the operation assessed are also conveyed to Unity for integration into the user interface. This interface comprises controls for adjusting the manufacturing parameters, which, in turn, feed information back into the Grasshopper algorithm. This reciprocal exchange of information enables the seamless configuration and

execution of the manufacturing process. For testing purposes, a graphic–user interface (GUI) was developed in Grasshopper and Unity with MRTK (Figure 9).

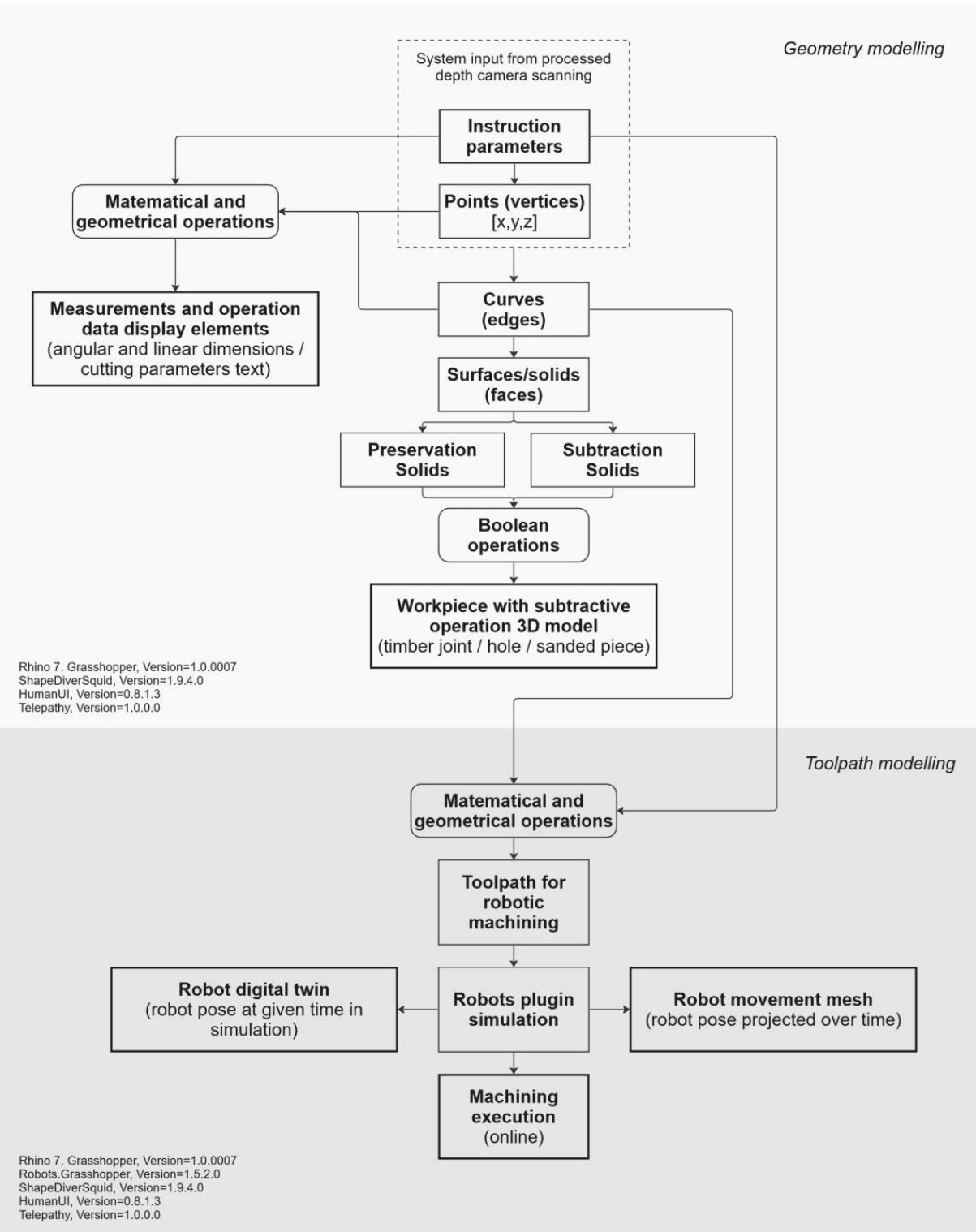

**Figure 8.** Geometric and toolpath modeling workflow.

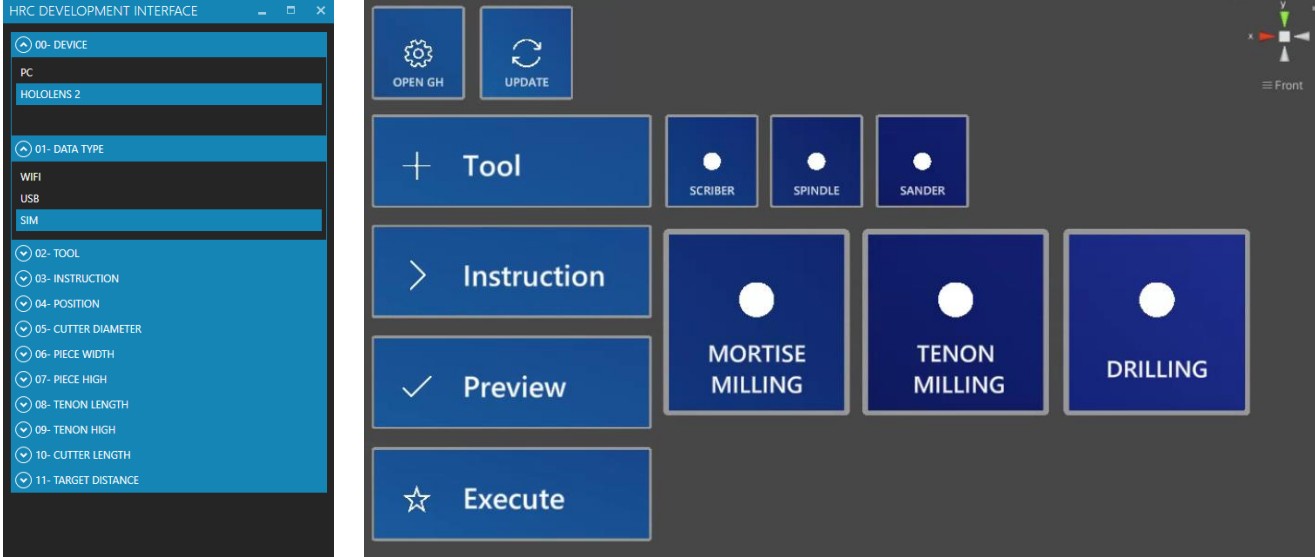

**Figure 9.** Testing GUI on Grasshopper (**left**) and Unity with MRTK (**right**).

A customized application for the Microsoft Hololens 2 device was compiled and deployed on the HMD as a Universal Windows Platform (UWP) application developed in Unity for data receiving from the computer running the Unity local application with Grasshopper (Figures 10–12).

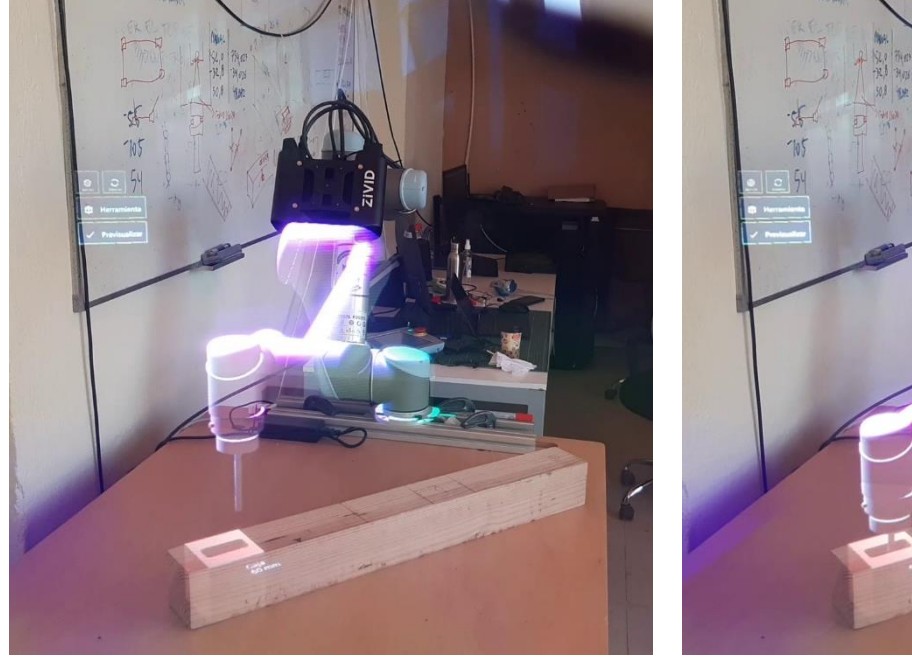
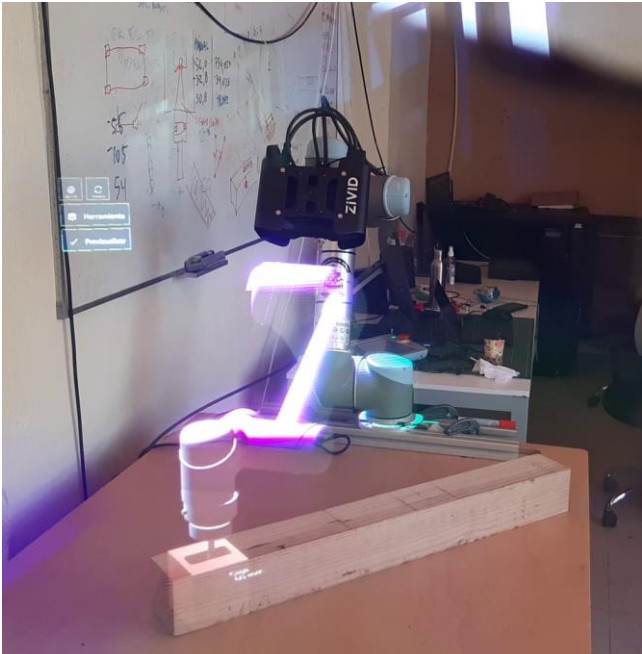

**Figure 10.** Operator point of view through MR-HMD; digital twin placement and operation.

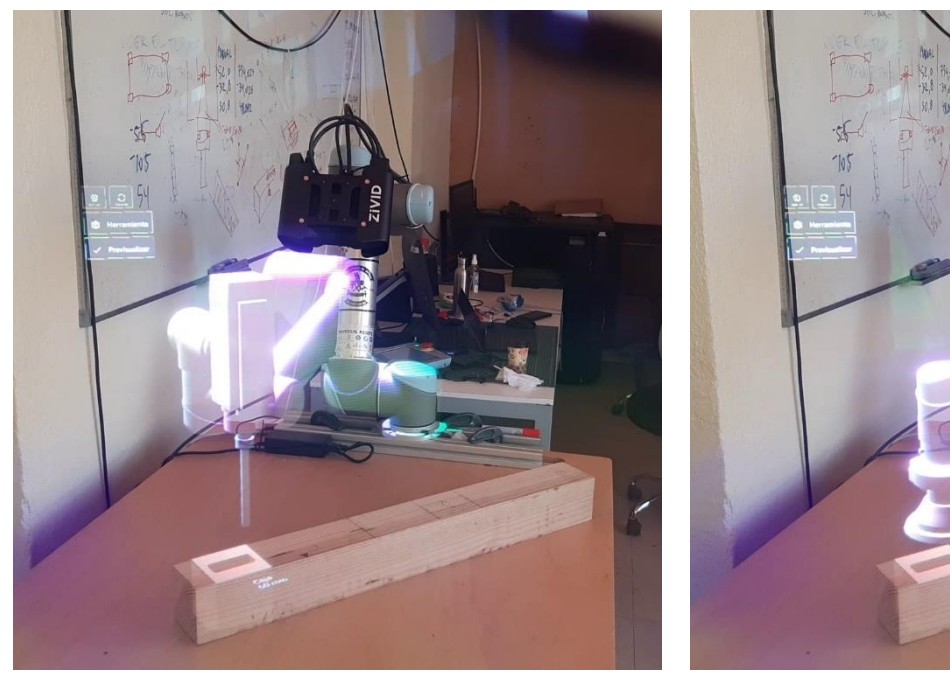
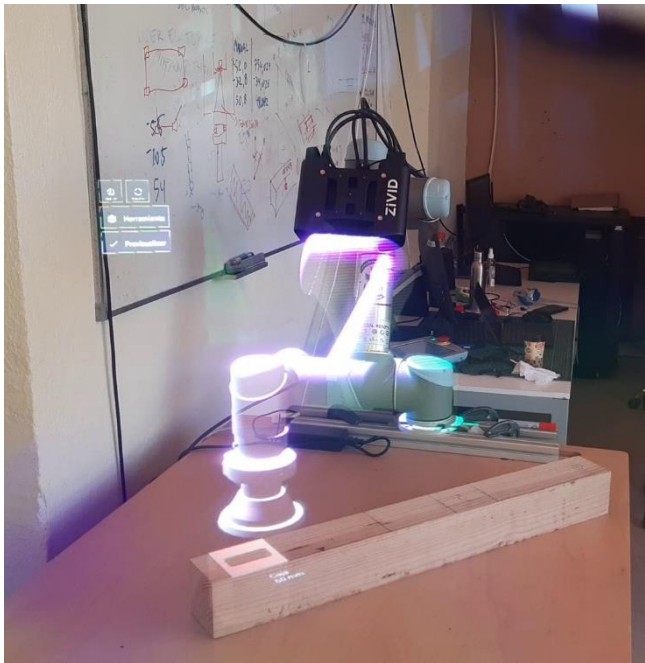

**Figure 11.** Operator point of view through MR-HMD; virtual tool change.

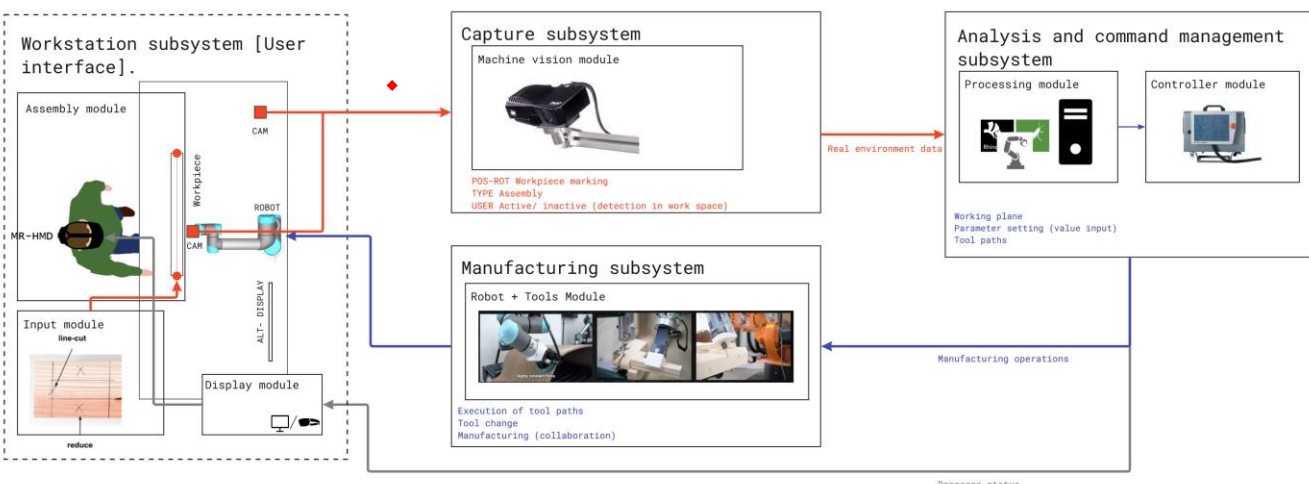

**Figure 12.** System architecture.

Since the robots' plugin in Grasshopper facilitates a real-time programming execution via its local connection with the UR5 robot controller, the integration of an activation parameter in the Unity interface enables the direct execution of the program in the real robot from the mixed–reality interface, placing and enabling an execution button.

### 3.4. Safety Awareness

The mixed reality system's visual feedback facilitates the real-time monitoring of the robot's behavior, aiding decision-making and accident prevention. It heightens the operator's awareness of potential hazards and unsafe conditions in the workspace, allowing for necessary safety measures such as repositioning the workpiece, halting the robot, or reorganizing the spatial layout, among others. In a setting where an unfamiliar robot collaborator is introduced, this visual feedback plays a critical role in educating operators about potential dangers, thus enabling proactive measures to prevent accidents and injuries.

However, it is still necessary to strengthen the system's precautionary measures and predictability capacity. For this reason, the Unity program incorporates a collision detection

system, testing the user's head and hands model provided by Hololens 2 tracking versus the projected volume of the robot's movement, which is constructed approximately by evaluating various stages of the manufacturing procedure and building a mesh. Real-time collision detection, using Unity's Mesh Collider components, triggers an alert that is displayed on the MR interface if the user is within the ur5 working area or in collision trajectory.

Using the manufacturer's description of the robot, the robot's working area is presented to the operator (Figure 13). In parallel, a collision volume is constructed, which enables its evaluation within Unity's local application with the mesh collider component when the user's hands or head cross the boundary of the working area. In such instances, a warning alert is displayed in the mixed–reality interface (a yellow warning icon) (Figure 14).

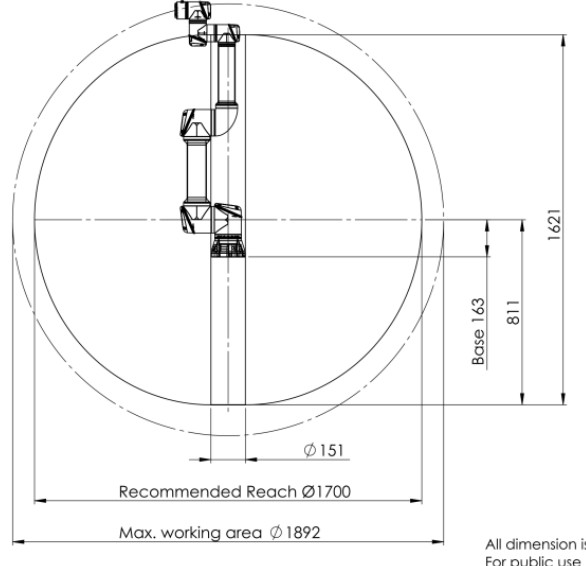

**Figure 13.** UR5 robot working area (dimensions in mm).

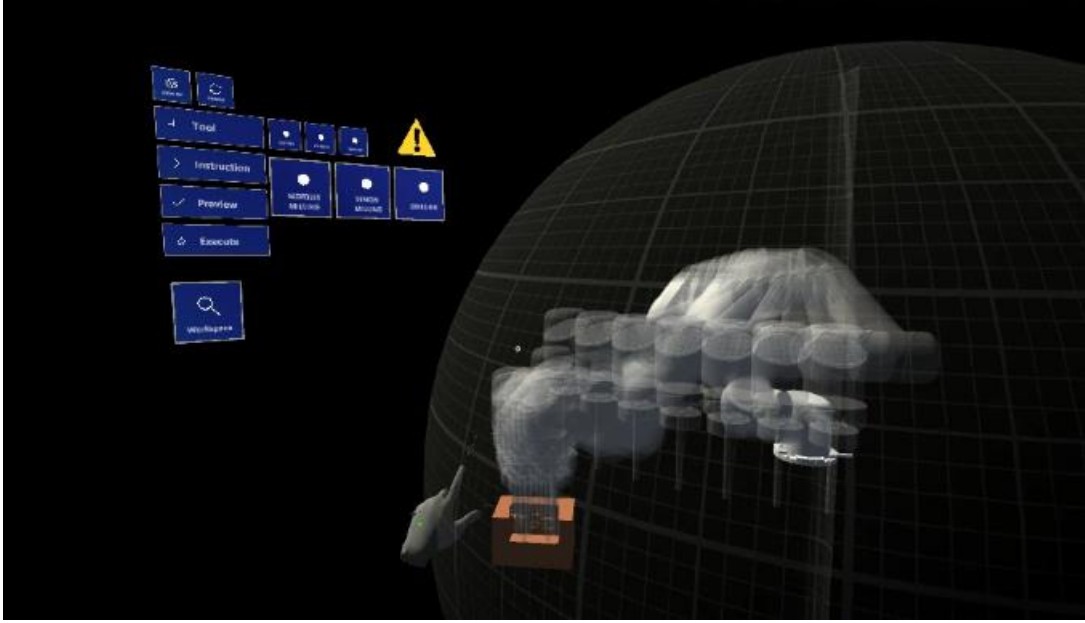

**Figure 14.** Collision detection system. UR5 working area trespassing warning.

Similarly, the programming developed in Unity allows for collision volumes to be assigned to the robot's movement path. This feature enables a real-time assessment of situations where the operator is at risk of direct collision with the robot. In instances where such a risk is detected, a warning alert is immediately displayed in the mixed–reality interface, taking the form of a red warning icon. This real-time warning system adds a critical safety feature, providing immediate visual feedback to the operator and enhancing the overall safety of human-robot interactions (Figure 15). Other types of warnings, such as those provided by the robot's plugin when evaluating tool paths, can also be taken into account and displayed on the same interface.

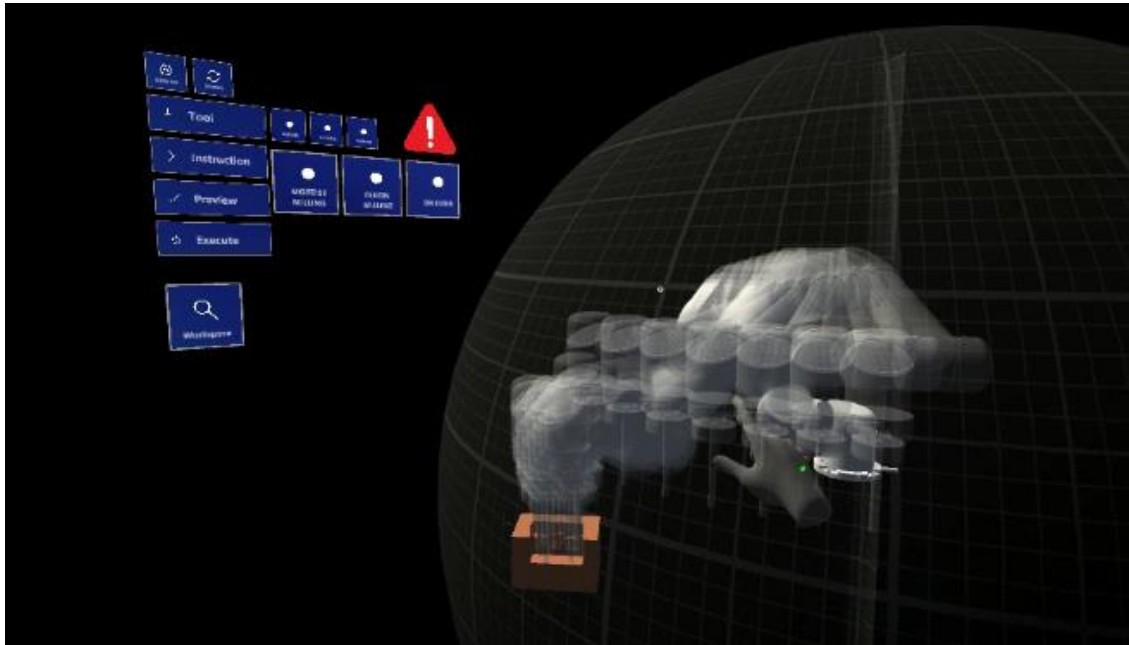

**Figure 15.** Collision detection system. UR5 collision warning on robot trajectory.

### 3.5. Experimental Procedure and Implementation

Based on this development, we propose three operations—Drilling, Mortise Milling, and Tenon Milling—to test the system. Each operation is evaluated in different positions on the workbench. This approach allows for an assessment of the robot's precision in describing tool paths, calculating collisions, and displaying information in each case. These operations must be exclusively controlled from the mixed–reality interface.

This proof of concept aims to evaluate the system based on minimal data points necessary for its operation (as shown in Figure 16). These input points vary depending on the type of instruction that is intended to be executed, which, at this time, must be selected in the Mixed–Reality (MR) interface.

Primarily, one or two recognized points, derived from the drawings on the workpiece, are needed to indicate the positioning on one of its faces, either for the drilling point or the cutting region (referenced as P1 and/or P2 in Figure 1). Subsequently, a point that signifies one of the edges of the workpiece is required (E1), which is marked at the intersection between a normal vector from one of the positioning points and a curve representative of the workpiece's edge. This allows the alignment of the instruction relative to the workpiece and creates a defined work plane to guide the tool. Lastly, at least one point on a plane normal to the work plane is required to indicate the depth and orientation of the cut into the workpiece (D1).

With these data, the system develops the geometric model, annotations, toolpaths, and robotic programming. The programming is evaluated by the robot's plugin in Grasshopper, which allows the operator to be informed whether it is correct or presents warnings or errors that prevent its execution. These are displayed in the mixed–reality interface to

correct the positioning of the workpiece to a position where the required instruction is feasible with the robot in use.

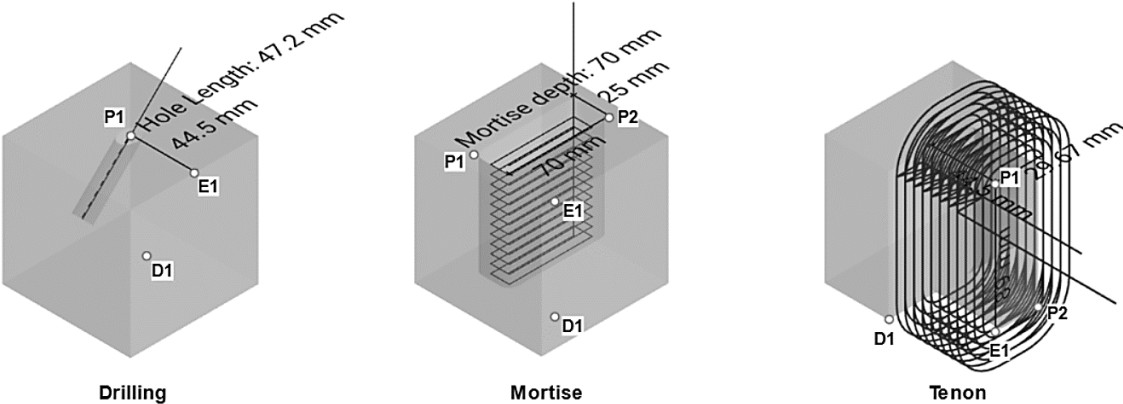

**Figure 16.** Input data required from simulated artificial vision.

After the preview model has been visualized and assessed, the operation is executed by the robot. While the final tools are not employed at this stage—just the steel scriber—it is vital to ensure the robot's movement and pose precisely match what is displayed in the preview (Table 1).

**Table 1.** Summary of performed experiments.

| Simulation | Workpiece Position | Input Data XYZ [mm] | Result |
|---|---|---|---|
| Drilling | Position 1 | P1: 595.92, 294.39, 49<br>E1: 622.28, 330.24, 49<br>D1: 647.1, 311.99, 13.24<br>Drill ⌀: 12 | • Target within reach.<br>• Correct toolpath generation.<br>• Operator presence detection in robot's workspace and toolpath.<br>• Drilling can be executed based on visual inspection.<br>• Hole length: 47.2 mm. |
| | Position 2 | P1: 347.88, 598.54, 49<br>E1: 392.38, 598.54, 49<br>D1: 392.38, 598.54, −5.8<br>Drill ⌀: 12 | • Target within reach.<br>• Correct toolpath generation.<br>• Operator presence detection in robot's workspace and toolpath.<br>• Drilling can be executed based on visual inspection.<br>• Hole length: 54.8 mm. |
| Mortise Milling | Position 1 | P1: 347.88, 598.54, 49<br>P2: 347.88, 598.54, 49<br>E1: 392.38, 598.54, 49<br>D1: 593.78, 467.44, −21<br>Cutter ⌀: 12 | • Target out of reach.<br>• Toolpath generation error.<br>• Operator presence detection in robot's workspace. "Target out of reach" warning display correctly.<br>• Mortise Milling cannot be executed on actual workpiece position.<br>• Mortise depth: 70 mm. |

**Table 1.** *Cont.*

| Simulation | Workpiece Position | Input Data XYZ [mm] | Result |
|---|---|---|---|
| | Position 2 <br>  | P1: 593.78, 410.47, 49 <br> P2: 504.78, 435.47, 49 <br> E1: 593.78, 467.44, 49 <br> D1: 593.78, 467.44, −21 <br> Cutter ⌀: 12 | • Target within reach. <br> • Correct toolpath generation. <br> • Operator presence detection in robot's workspace and toolpath. <br> • Mortise Milling can be executed based on visual inspection. <br> • Mortise depth: 70 mm. |
| Tenon Milling | Position 1 <br>  | P1: 458.38, 477.37, 49 <br> P2: 435.58, 458.39, −40 <br> E1: 458.38, 477.37, −40 <br> D1: 509.57, 462.08, −40 <br> Cutter ⌀: 12 | • Target within reach. <br> • Correct toolpath generation. <br> • Operator presence detection in robot's workspace and toolpath. <br> • Tenon Milling cannot be executed. The tool may hit the table; detection is through visual inspection by the operator. <br> • Tenon length: 44.5 mm. |
| | Position 2 <br>  | P1: −228.35, 261.28, 49 <br> P2: −228.35, 231.61, −40 <br> E1: −228.35, 261.28, −40 <br> D1: −183.85, 290.83, −40 <br> Cutter ⌀: 12 | • Target within reach. <br> • Correct toolpath generation. <br> • Operator presence detection in robot's workspace and toolpath. <br> • Mortise Milling can be executed based on visual inspection. <br> • Tenon length: 44.5 mm. |

## 4. Results

The conducted tests have confirmed various phenomena associated with the human-robot collaboration using the proposed system. This includes a straightforward yet effective alert system that is triggered upon the potential collision between representative volumes of the operator (specifically the hands and head) and the robot (its physical structure and maximum reach volume). Therefore, when working near the robot within the collaborative workspace provides essential safety feedback for the operator (Figure 17). This safety measure is further enhanced by the visual inspection of digital elements within the real environment, helping to prevent collisions with objects or other obstructions, such as the worktable, which is one example of elements that are not yet integrated into the sensor system. The robot's full range of motion, represented in multiple instances, can be displayed through the mixed-reality headset. This geometry serves as a method for evaluating potential collisions throughout the robot's movement during each operation, although it is not necessary to constantly display it (Figure 18).

This safety system operates efficiently across all trajectories that are successfully generated. If issues arise, such as problems with the robot's joints due to the programmed pose or errors caused by targets being out of reach, the system triggers corresponding alerts. These alerts are transmitted from the log of the robot's plugin in Grasshopper. This informs the operator and allows for the repositioning of the workpiece as well as the reassessment and potential generation of new toolpaths (Figure 19).

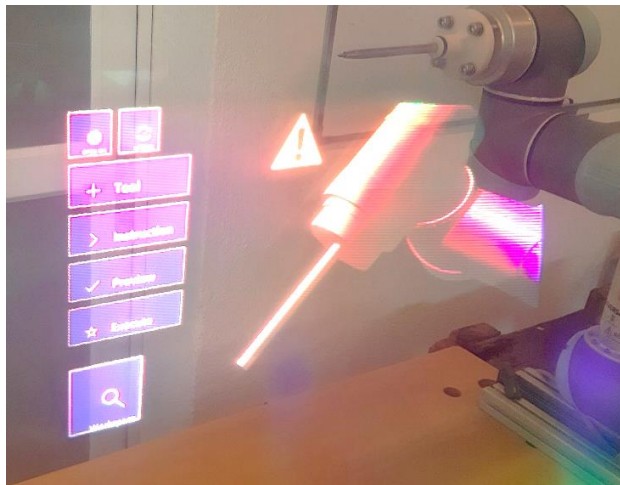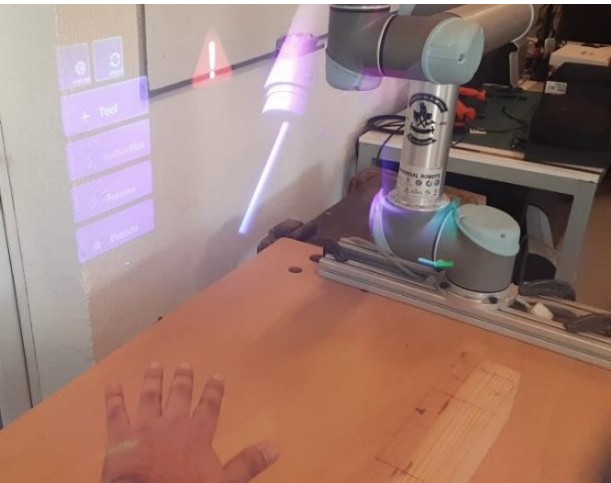

**Figure 17.** (**Left**) Visual alert indicating crossing into the robot's maximum reach volume; (**Right**) Detection of operator in collision path with the previewed toolpath.

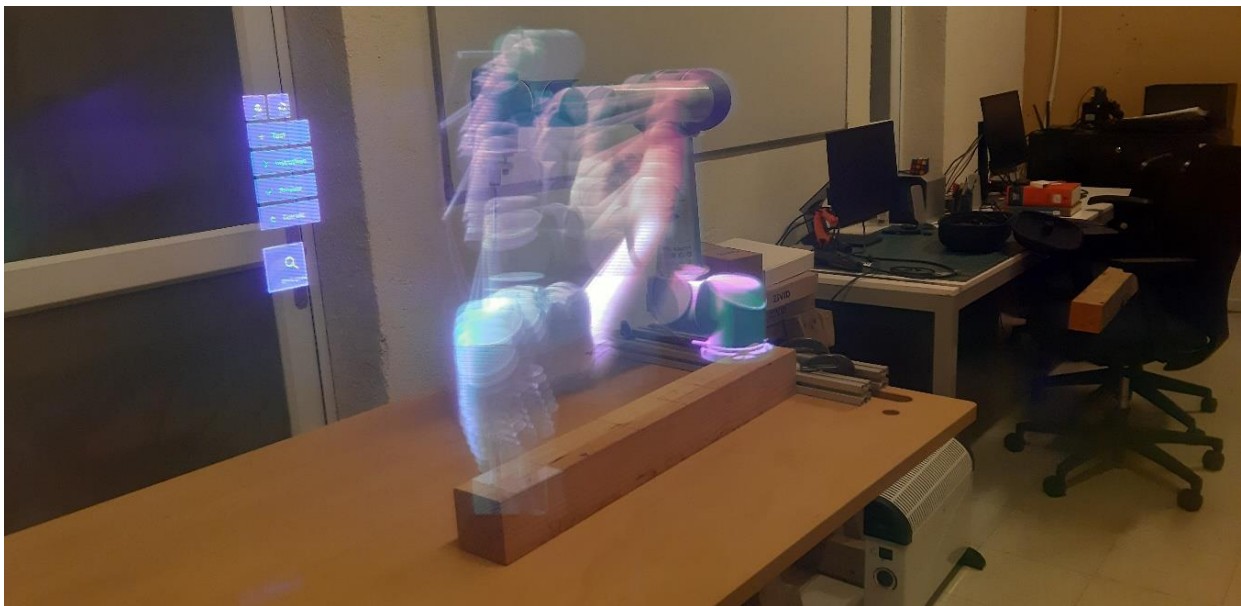

**Figure 18.** Geometry representing the robot's full range of motion at a given instruction, displayed through the mixed reality headset.

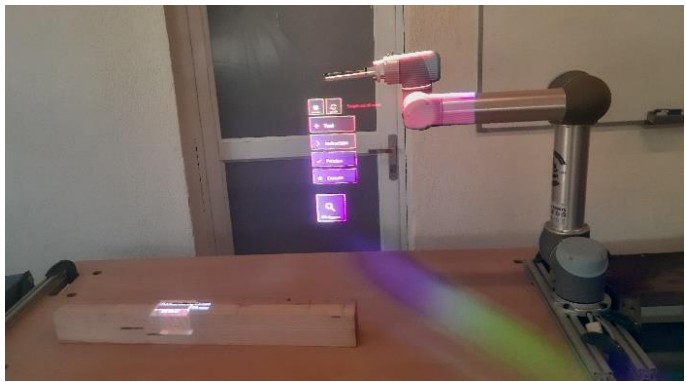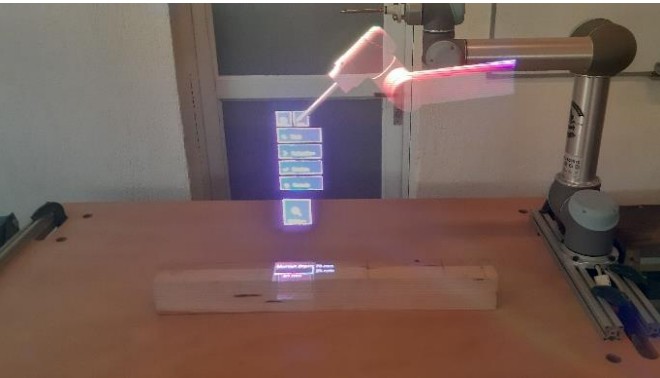

**Figure 19.** (**Left**) Target out of reach; (**Right**) Target within reach of the robotic arm.

The arrangement of visual guides effectively communicates the implications of the instructions provided through hand-drawn symbols on the workpiece. However, positioning errors and broad tolerance have been observed. The system does not achieve high precision in integrating the digital and the virtual realms, but it is adequate for preventing collisions and making workpiece positioning corrections for robotic manufacturing. The assessment of a drawn instruction's position and the development of trajectories work well as long as the reading of the symbols and identification of the required points are precise: a factor reliant on the effectiveness of computer vision and machine learning in ongoing research by the same research team. Thus, even if the positioning of the visual guides on the MR-HMD does not achieve high precision, tool positioning reliability relative to the robotic workpiece depends solely on the artificial vision system's reading. It is crucial to understand that while achieving maximum precision is not vital for visual feedback, it is indispensable for the robot's artificial vision system, which detects the targets and ultimately positions the tool.

The primary contribution of the visual guidance system is the verification and understanding of the instructions given by hand-drawn symbols that aid in decision-making regarding the positioning of the workpiece and other elements within the collaborative workspace within an approximate range. The system, in its entirety, is versatile and allows for positioning the workpiece within a wide range of possibilities. The information provided eases the human contribution to ensure successful collaboration with the robot, as it aids in comprehending the spatial implications of its use, which the proposed system effectively achieves.

In some instances, the significant displacement of the visual guides on the workpiece can be observed when the operator's initial viewing angle and its position vary drastically (Figure 20). This represents a major drawback in current development. Addressing this issue within the framework of the Windows Mixed Reality Toolkit (MRTK) in Unity could involve potential enhancements such as incorporating workspace markers or other auxiliary guides to ensure the accurate positioning of instructions feedback.

Operations simulated and evaluated with the system are accurately reflected in the robot's movement for a given instruction. Once the cutting positions are input into the workpiece, the system successfully displays tool paths and the volume through which the robot can move. It also provides real-time warnings if any part of the operator's body is at risk of collision.

The evaluation of various simulated manufacturing processes using the proposed cyber–physical system underscores the adaptability of human-robot collaboration (HRC) in unstructured environments. Through testing diverse workpiece positions (Figure 21), we have demonstrated the feasibility of developing suitable trajectories and effectively mediating the spatial implications of the manufacturing process. This helps foster enhanced communication and mutual trust between the operator and robot, bolstering safety for both the operator and other users within the workspace. Notably, empowering the operator to inform others about the robot's behavior further reinforces the safety aspect of this implementation.

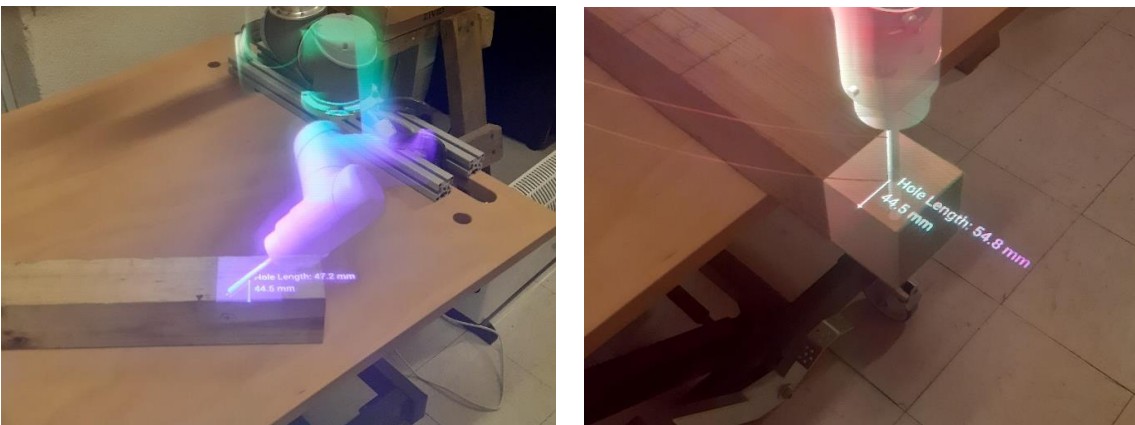

**Figure 20.** Tenon milling preview through MR-HMD. Displacement of visual guidance.

**Figure 21.** *Cont.*

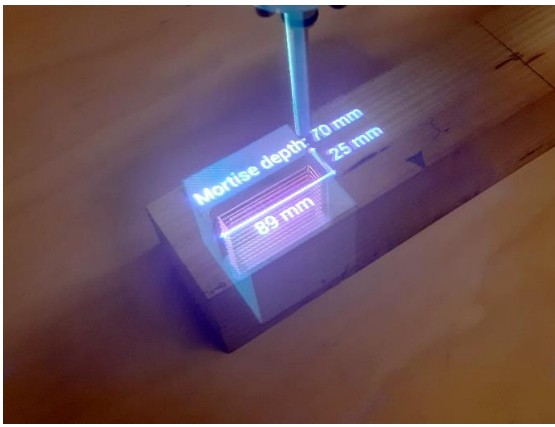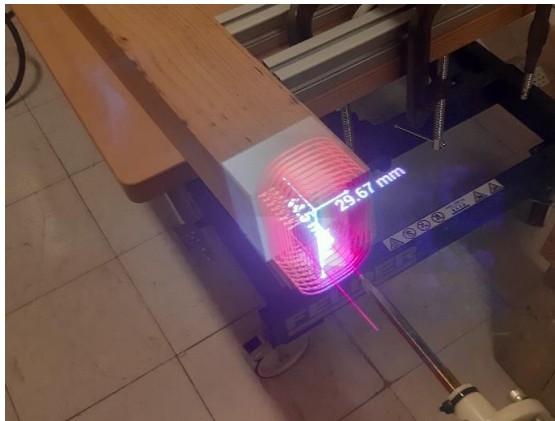

**Figure 21.** Display of different positioning results for visual guides, as viewed through the MR-HMD, HoloLens 2.

## 5. Discussion

Typically, new users of industrial robots must learn and become accustomed to advanced manufacturing techniques, which requires knowledge of programming and digital modeling in specialized software. However, this research contributes to a workflow that incorporates traditional timber construction approaches, where the robotic system adapts to human actions, is proprioceptive, and explicitly communicates its behavior, avoiding large abstractions or reinterpretations. This approach could counter the potential threat of automation to traditional trades like carpentry by shifting the focus from humans adapting to robots to robots adapting to humans, indicating a paradigm shift.

We anticipate that this approach can fortify the carpentry industry, benefiting both present and future carpenters by facilitating their integration into the dominant production model of Industry 4.0. Suitable for collaborative robotics, including robots with six or more degrees of freedom, this method holds the potential to redefine traditional carpentry workspaces. Its flexibility in workpiece positioning, along with the diversity of tools that can be employed within the same robotic unit, present significant advantages.

This workflow eliminates the need to establish a "zero part" or reference point in the CNC system, as it is automatically detected through user signaling on the workpiece, allowing for multiple positions or orientations. By combining multiple readings of the symbols on the workpiece into a manufacturing sequence, which is still under development, a completely new and customizable wood joinery with one or more operators can be created. The accumulation of operations and their outcomes can be evaluated in advance using Mixed Reality (MR), allowing for potential adjustments and improvements.

Optical see-through head-mounted devices, such as the Hololens 2, leverage Mixed Reality (MR) to provide real-time, stereoscopic visualizations of the robot's intended actions in the actual environment. This immediate feedback enhances the human operator's understanding of robotic behavior through direct experience. Through MR overlays on the robot and the environment, operators can visualize the robot's planned path, task progress, and even future movements. This approach builds trust by enabling humans to anticipate the robot's actions and intentions, checking communication effectiveness between the human and robot. The MR system further promotes safety by highlighting danger zones and displaying warnings when the robot is about to intrude into the operator's workspace. However, for this system to function with enhanced efficiency, it is necessary to incorporate the more precise tracking of additional elements within the workspace into the predictive collision evaluation system. As it stands, the current system only enables automated detection of potential collisions between the robot and the operator. Furthermore, optimizing the existing data flow to improve the system's response time is an essential next step.

By utilizing mixed reality technologies, industrial environments can augment human-robot interactions, establish effective communication channels, and cultivate mutual trust. The amalgamation of visualization, safety enhancements, remote collaboration, interactive training, a contextual information display, and intuitive interfaces can contribute to more transparent, efficient, and reliable human-robot interaction.

The versatility of the collaborative robot opens up possibilities for a mobile workstation that is significantly smaller than a traditional workshop. Future work includes a detailed definition of the shared spaces between humans and robots, the optimization of auxiliary trajectories, and the fine-tuning of tool orientations to accommodate these shared spaces. This involves strategizing for optimal positioning and orientation in various working environments and coexisting with other workflow processes involved in building construction.

Future evaluations involving various carpenters are essential when assessing the proposed performance parameters and user experience. Though the system is still in its early stages and requires the detailed calibration of assembly tools, it demonstrates functionality and shows potential for effective manufacturing. Future tests could not only assess the efficiency and effectiveness of this method but also compare it with the current state of automated development in terms of design scope, customization possibilities, and adaptability, with the aim of achieving significant advantages. According to ongoing research, it is possible to redefine the input information, thereby enabling a more specific and customizable process for the definition of complex wood assemblies with various shapes and angles through an accumulative sequence of instructions.

For the effective implementation of this system with collaborative robots, payload constraints and, therefore, the mechanical capabilities of the system must also be addressed as these could lead to procedural faults. Vysocky et al. [4] pointed out that payload is one of the primary concerns for companies evaluating the use of these systems. On the other hand, the current method allows the integration of various types of robots; as the offerings of collaborative robots are aligned more closely with industry needs, it becomes possible to integrate this workflow.

## 6. Conclusions

This research focused on the development of a cyber–physical system enabling human-robot collaboration (HRC) for wood stereotomy with instructional visual language and visual guidance, providing a contribution to the timber construction industry. In the context of Construction 4.0 and advances in automation, human-robot collaborations represent an opportunity for integration. The proposed system, which adapts robots to human actions, has the potential to greatly facilitate and accelerate training and technology adoption processes. Using HRC assisted by MR, it is possible to improve the synergy between humans and robots, leading to increased efficiency, productivity, and overall performance in the construction sector. This research provides a solid foundation for the further development and implementation of HRC systems in woodworking, focusing on safety awareness and reliability in variable tasks and working environment conditions.

The implementation of this system could lead to improvements in the workspace planning process. By swiftly and reliably understanding the robot's behavior, the operator is empowered to make decisions regarding the arrangement of objects in space, particularly in instances where flexibility in manufacturing or adaptation to variable and narrow spaces is desired. This capability provides agile and trustworthy insights that assist the operator in optimizing the organization of the workspace. The system's ability to support decision making in workspace planning enhances efficiency, adaptability, and productivity in manufacturing operations, ultimately benefiting the overall manufacturing process.

**Author Contributions:** Conceptualization, E.V.-A. and L.F.G.-B.; methodology, E.V.-A. and L.F.G.-B.; software, E.V.-A.; validation, E.V.-A.; formal analysis, E.V.-A. and L.F.G.-B.; investigation, E.V.-A. and L.F.G.-B.; resources, E.V.-A.; data curation, E.V.-A.; writing—original draft preparation, E.V.-A. and L.F.G.-B.; writing—review and editing, E.V.-A. and L.F.G.-B.; visualization, E.V.-A.; supervision, L.F.G.-B.; project administration, L.F.G.-B.; funding acquisition, L.F.G.-B. All authors have read and agreed to the published version of the manuscript.

**Funding:** The research leading to these results has received funding from the project titled "Robotic timber joinery for onsite prefabrication of flexible housing solutions at reconstruction processes led by SERVIU-MINVU" in the frame of the program "The Scientific and Technological Development Support Fund (FONDEF)" managed by the Chilean National Commission for Scientific and Technological Research (CONICYT), under the Grant agreement number FONDEF ID20I10262".

**Data Availability Statement:** Not applicable.

**Acknowledgments:** We are deeply grateful to the colleagues of the AReA (Área Robots en Arquitectura) laboratory of the Department of Architecture at the Universidad Técnica Federico Santa María.

**Conflicts of Interest:** The authors declare no conflict of interest. The funders had no role in the design of the study; in the collection, analyses, or interpretation of data; in the writing of the manuscript; or in the decision to publish the results.

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
