# Peer review of "Mixed Reality for Safe and Reliable Human-Robot Collaboration in Timber Frame Construction"

_buildings, doi:10.3390/buildings13081965_

Round 1

Reviewer 1 Report

Dear authors

The following comments I suggest to consider in order to improve your paper

1)    The aim of the paper is not clear. Please rewrite the abstract. I suggest using the following structure:

a.     The context in the field of study

b.     Problems and how are solved

c.     The gap

d.     How does this research cover this gap

e.     A short mention of the obtained results

Moreover, the introduction section may follow the same structure, but with more details.

2)    Citation number along the paper may be increased, I suggest 2x (For instance, the following sentences should be supported by a citation):

a.     Lines 31-32

b.     Lines 32-25

c.     Lines 40-42

d.     Lines 42-44

e.     Lines 68-71

f.      Lines 100-102

g.     Lines 137-138

h.     Line 150

3)    Some sentences are too long to be understood in English. The following sentences should be split:

a.     Lines 81-85

b.     Lines 86-93

c.     Lines 110-118

4)    Figure 8 should be presented in English. Some buttons on the screen are not readable, for instance “lijadora?” “marcador”?

5)    Results should be improved. How the performed simulation may be described through a graph? Diagram? The results in terms of values? Successful instances? Not successful instances?

Author Response

Response to Reviewer 1 Comments

Point 1: The aim of the paper is not clear. Please rewrite the abstract. I suggest using the following structure:

  1. The context in the field of study
  2. Problems and how are solved
  3. The gap
  4. How does this research cover this gap
  5. A short mention of the obtained results

Moreover, the introduction section may follow the same structure, but with more details.

Response 1: We have considered the reviewer's suggestions and have rewritten the abstract accordingly.

Point 2: Citation number along the paper may be increased, I suggest 2x (For instance, the following sentences should be supported by a citation):

  1. Lines 31-32
  2. Lines 32-25
  3. Lines 40-42
  4. Lines 42-44
  5. Lines 68-71
  6. Lines 100-102
  7. Lines 137-138
  8. Line 150

Response 2: In response to the reviewer's comments, we have added additional references in the specified lines. This has increased the number of citations in our manuscript, thereby improving its comprehensiveness and relevance to the field of study

Point 3: Some sentences are too long to be understood in English. The following sentences should be split:

  1. Lines 81-85
  2. Lines 86-93
  3. Lines 110-118

Response 3: We have divided the sentences identified by the reviewer into shorter, more manageable segments. Furthermore, we have taken the initiative to revise other sentences within the manuscript that presented the same issue. We believe these changes enhance readability and clarity throughout the paper.

Point 4: Figure 8 should be presented in English. Some buttons on the screen are not readable, for instance “lijadora?” “marcador”?

Response 4: Now, it is presented in English and text is changed.

Point 5: Results should be improved. How the performed simulation may be described through a graph? Diagram? The results in terms of values? Successful instances? Not successful instances?

Response 5: Due to last-minute technical issues with the software and the time constraints to respond to the request, a detailed record of the experiments performed is pending. We understand the necessity and urgency to articulate the obtained results in a better manner. Generally, the results are satisfactory, however, they should be recorded in detail.

Reviewer 2 Report

The authors propose a human robot collaboration system with visual feedback from a robotic manipulator to be used in wood stereotomy. They defined a clear workflow where the worker positions the wooden piece and the robot cuts it. A computer vision system was implemented to detect markings on the piece, presenting them through their augmented reality system for validation. Based on the interpreted markings, their robot will perform specific motions and these can be previsualized through their augmented reality system.

The article is well written and is an interesting implementation of a HRC system. Nevertheless there are a few aspects that should be improved:

1) Schematics should be in english;

2) Explain in greater detail the collision detection system, in particular how the hands and head tracking data is being integrated in Unity; Place a few print screens illustrating how this information is presented to the operator; Being a digital twin, this tracking data should be displayed in real time on the virtual environment too, not just the robotic arm current and planned positions, as well as the work piece.

3) Explain the planning methodology used for executing the tasks with the robotic manipulator, considering the obstacles in the workspace.

4) The results are lacking:

- Need experiments (different scenarios/workpiece configurations/markings) that demonstrate that the pipeline (from workpiece detection, marking identification, mixed reality feedback, to the execution) is robust.

- What is the feedback on the mixed reality interface from the marking detection phase? Specify all markings and illustrate the recognized through the AR system.

- The mixed reality platform is described and illustrated but the alert notifications to the worker were not shown.

Best regards

Author Response

Response to Reviewer 2 Comments

Point 1: Schematics should be in english;.

Response 1: Now the Schematics are in english.

Point 2: Explain in greater detail the collision detection system, in particular how the hands and head tracking data is being integrated in Unity; Place a few print screens illustrating how this information is presented to the operator; Being a digital twin, this tracking data should be displayed in real time on the virtual environment too, not just the robotic arm current and planned positions, as well as the work piece.

Response 2: We have deepened in the collision system programmed in Unity. There are still details to be commented, regarding the amount and types of information that the system allows to display to the operator.

Point 3: Explain the planning methodology used for executing the tasks with the robotic manipulator, considering the obstacles in the workspace.

Response 3: This point is not yet fully addressed in the manuscript, we consider it to be relevant information, which we possess but have not yet integrated into the text. Planning regarding the workspace develops as a traditional carpenter would naturally do with their equipment and work environment, now adding information about the space and time usage that the robot implies. This allows decision-making, both in the location and relocation of workpieces in consideration of the visual feedback, as well as of the elements and other workers present in the place, through simple visual inspection

Point 4: The results are lacking:

 - Need experiments (different scenarios/workpiece configurations/markings) that demonstrate that the pipeline (from workpiece detection, marking identification, mixed reality feedback, to the execution) is robust.

 - What is the feedback on the mixed reality interface from the marking detection phase? Specify all markings and illustrate the recognized through the AR system.

 - The mixed reality platform is described and illustrated but the alert notifications to the worker were not shown.

Response 4: Due to last-minute technical issues with the software and the time constraints to respond to the request, a detailed record of the experiments performed is pending. We understand the necessity and urgency to articulate the obtained results in a better manner. Generally, the results are satisfactory, however, they should be recorded in detail. We have displayed a table with the experiments performed, but the specific data are still pending. Now showing some of the feedback that allows the system to deliver in Unity, images of on-site usage through the viewers can be added.

Round 2

Reviewer 1 Report

A formal response has not been received.

Comments on point 5) of the previous revision have not been resolved or answered.

When a new revision is resubmitted, I suggest presenting a document responding to each reviewer's comment and the location of the improvements made to the manuscript.

Author Response

Response to Reviewer 1 Comments

Thank you for the feedback provided and the comments, the response to each point is presented below.

Point 1: A formal response has not been received.

Response 1: We previously utilized the 'Author's Notes to Reviewer' section in round 1 on the webpage and now we have attached a PDF backup for reference. The updated manuscript .docx includes tracked changes for review.

Point 2: Comments on point 5) of the previous revision have not been resolved or answered

Response 2:  We are referencing point 5 from the comments in round 1 of the review:

“Results should be improved. How the performed simulation may be described through a graph? Diagram? The results in terms of values? Successful instances? Not successful instances?”

Now the manuscript addresses this comment by presenting the experiments conducted with the system, evaluating different positions for collaborative manufacturing, and the results in the visual feedback provided.

Reviewer 2 Report

The authors made some of the suggested changes, in particular related to the first 2 points mentioned in the previous round. The changes are positive and improve the quality of the paper. Nevertheless, since some of them are still pending, I'll maintain my recommendation on the article's acceptance for now. MDPI's journals are always in a hurry to receive the updated articles, so I understand that time is always short. I look forward to receive the fully updated article in the next round. 

English is fine. I detected this sentence on page 11 that needs to be changed: "Since the Robots plugin in Grasshopper facilitates real-time programming execution via a local connection with the UR5 robot controller."

Author Response

Response to Reviewer 2 Comments

Point 1: The authors made some of the suggested changes, in particular related to the first 2 points mentioned in the previous round. The changes are positive and improve the quality of the paper. Nevertheless, since some of them are still pending, I'll maintain my recommendation on the article's acceptance for now. MDPI's journals are always in a hurry to receive the updated articles, so I understand that time is always short. I look forward to receive the fully updated article in the next round.

Response 1: Now the manuscript addresses this comment by presenting the experiments conducted with the system, evaluating different positions for collaborative manufacturing, and the results in the visual feedback provided. This includes an in-depth explanation of the input data to the system concerning the symbols on the workpieces. However, the language of symbols itself is the subject of another ongoing research project that delves into its effectiveness and the variety required to generate different types of wood assemblies.

Round 3

Reviewer 2 Report

Overall it is a very interesting paper, congrats!